# Acute Effects of Tecar Therapy on Skin Temperature, Ankle Mobility and Hyperalgesia in Myofascial Pain Syndrome in Professional Basketball Players: A Pilot Study

**DOI:** 10.3390/ijerph18168756

**Published:** 2021-08-19

**Authors:** Mireia Yeste-Fabregat, Luis Baraja-Vegas, Juan Vicente-Mampel, Marcelino Pérez-Bermejo, Iker J. Bautista González, Carlos Barrios

**Affiliations:** 1Doctoral School, Catholic University of Valencia (UCV), 46001 València, Spain; 2Department of Physiotherapy, Faculty of Medicine and Health Sciences, Catholic University of Valencia (UCV), 46001 València, Spain; luis.baraja@ucv.es (L.B.-V.); juan.vicente@ucv.es (J.V.-M.); ikerugr@gmail.com (I.J.B.G.); 3Department of Biostatistics, Epidemiology and Public Health, Faculty of Medicine and Health Sciences, Catholic University of Valencia, 46001 València, Spain; marcelino.perez@ucv.es; 4Institute for Research on Musculoskeletal Disorders, Catholic University of Valencia, 46001 Valencia, Spain; carlos.barrios@ucv.es

**Keywords:** trigger point, sport injuries, gastrocnemius muscle, diathermy, thermography, basketball, range of motion

## Abstract

(1) Background: Myofascial pain syndrome (MPS) is a clinical condition characterized by localized non-inflammatory musculoskeletal pain caused by myofascial trigger points. Diathermy or Tecar therapy (TT) is a form of noninvasive electro-thermal therapy classified as deep thermotherapy based on the application of electric currents. This technique is characterized by immediate effects, and its being used by high performance athletes. (2) Methods: A total of thirty-two participants were included in the study who were professional basketball players. There was a 15-person Control Group and a 17-person Intervention Group. TT was applied in the Intervention Group, while TT with the device switched off (SHAM) was applied in the Control Group. The effects were evaluated through the Lunge test, infrared thermography, and pressure threshold algometry at baseline, 15, and 30 min after the intervention. (3) Results: the Intervention Group exhibited a greater increase in absolute temperature (F_[1,62]_ = 4.60, *p* = 0.040, η^2^_p_ = 0.13) compared to the Control Group. There were no differences between the groups in the Lunge Test (F_[1.68,53.64]_ = 2.91, *p* = 0.072, η^2^_p_ = 0.08) or in pressure algometry (visual analog scale, VAS) (F_[3.90]_ = 0.73, *p* = 0.539, η^2^_p_ = 0.02). No significant short-term significant differences were found in the rest of the variables. (4) Conclusions: Diathermy can induce changes in the absolute temperature of the medial gastrocnemius muscle.

## 1. Introduction

Electrical or electromagnetic stimulation-based physical therapy has been applied in rehabilitation with successful results. Specifically, resistive capacitive electrical transfer therapy has been used in physical rehabilitation and sports medicine to treat muscle, bone, ligament, and tendon injuries [1,2]. Radio frequency energy is currently the most commonly used energy source to generate therapeutic heat levels [2] in injuries related to muscle stiffness [2].

Tecar therapy (TT) is a form of non-invasive electrothermal therapy classified as deep thermotherapy based on the administration of electric currents within the radiofrequency range, constituting a monopolar capacitive resistive radiofrequency of 448 KHz [1,3,4,5]. This technique is based on the use of high frequency electromagnetism (less than 3 MHz) [6]. TT is characterized by its speed of action, so it is used in high performance sports [4], as this tool accelerates the recovery process [7]. Thermal changes produced by TT within the neuromuscular structure induce vasodilation, reduce muscle spasms, accelerate cellular metabolism, and increase soft tissue extensibility [8,9].

Myofascial pain syndrome (MPS) is a clinical condition characterized by localized non-articular musculoskeletal pain [10] as a consequence of myofascial trigger points (MTrPs) [11] located in the muscle [12,13,14]. MPS is considered one of the most common causes of muscle pain [11]. A total of 85% of patients with chronic pain suffer from MPS [15]. Latent MTrPS can cause motor dysfunction such as stiffness, restricted range of motion, and muscle fatigability, but it did produce not spontaneous sensory symptoms unless stimulated by pressure [16].

MPS is caused by trauma or muscle overuse in certain sports or activities. It can also appear in a weak muscle when demands exceed capacities in terms of the muscle activity not being able to withstand the strain exerted [17]. In addition, latent MTrPs reduce the joint range of motion due to muscle shortening from muscle and tendon stiffness [18].

Travell and Simons used the term myofascial trigger points (MTrPs) [11], characterizing them as the most sensitive palpable nodule in a taut band of skeletal muscle [3,14]. They are hyperirritable points that in the presence of pressure, stimulate MPS, inducing local, referred pain and stimulus responses [14]. Referred pain was described as a constant, deep, and intense pain, that is reproducible and predictable [19]. From a clinical perspective, there are two types of MTPs, namely active or latent. On the one hand, active MTPs produce sensory symptoms and motor dysfunction (i.e., restriction of movement and decreased range of motion (ROM)). Meanwhile, latent MTPs are able to modify muscle contraction without sensory symptoms, unless they are stimulated manually. The clinically evident progression from a nontender taut band to a tender band suggest a change in the muscle, signifying the development of an MTP [16]. If active MTPs are considered as peripheral nociceptive sources that are able to maintain a central sensitization state, latent MTPs have the same characteristics with a lower degree of sensitization (i.e., they do not manifest spontaneous pain) [16]. For instance, several research studies have shown that latent MTPs restrict ankle ROM [17,20].

Triceps sural injuries are common in a wide number of sports but are mainly prominent in soccer [21,22]. There are several studies based on diagnostic imaging [23] through musculoskeletal ultrasound, which indicate that the most prevalent injuries take place in the medial gastrocnemius (58–65%) [24]. The gastrocnemius muscle was chosen for this study because it has been shown to exhibit the highest prevalence of latent MTrPs in healthy participants [25]. During the examination process, physiotherapists may use referred pain patterns to understand and recognize how the MPS developed [19].

However, the acute impact of the TT therapeutic tool on the treatment of medial gastrocnemius related to MTP has not yet been investigated. Thus, the main aim of the present study was to analyze the acute effect of TT on latent MTPs on skin temperature, ankle ROM, and pain in professional basketball players. We hypothesized that the use of TT would produce an acute increase of skin temperature and ankle ROM and, at the same time, cause a decrease of hyperalgesia associated with MTPs.

## 2. Materials and Methods

### 2.1. Design

This is a randomized clinical trial involving thirty-two participants. Once the participants completed the preliminary stages, they were randomized using a simple Excel procedure in two groups: The Diathermy Group and Control Group. The process described above was managed by a staff member who did not participate in the study. All research staff were blinded to this allocation process. Single-blinding was used to reduce bias in the interpretation of the results. The patients were not informed of the treatment being performed in the immediate post-treatment and successive measurements.

### 2.2. Sample

A total of thirty-two (*n* = 32) amateurs basketball players were involved in the present study coming from the Picanya national category basketball team (Valencia). The mean ± SD of age, height, body mass, and body mass index were 22.84 ± 5.86 years; 179.33 ± 7.98 cm; 75.73 ± 11.51 kg; and 23.51 ± 2.81 kg/m^2^. All of the participants were previously informed about the procedure and signed the informed consent prior to their participation. A parental or guardian’s authorization and consent form was attached for the participants who below 18 years of age. The research was previously registered in ClinicalTrials.gov (ID: NCT04325750—https://clinicaltrials.gov/ct2/show/NCT04325750 (23 March 2020).

The following exclusion criteria were applied: (a) connective tissue pathology; (b) lymphatic disorders (lymphadenopathy); (c) skin injuries (open wounds, infection, psoriasis, tattoos, hematoma); (d) peripheral neuropathies; (e) previous fractures; (f) previous lower limb surgeries (in the past 12 months). There were three adverse events during the investigation process (post-treatment skin reactions) resulting in a total of three dropouts who were excluded from the initial sample.

The inclusion criteria were: (a) be an active national-level male basketball player; (b) be aged between 16–39 years of age; (c) not having suffered leg injuries in the past 6 months; (d) not having suffered a triceps sural rupture in the past two years; (e) show a difference of 1.5 cm between limbs in ankle dorsiflexion restriction measured through the lunge test; (g) be diagnosed with latent MTPs in the gastric–soleus complex through manual therapy in the dominant side.

Latent TrPs were identified as follows: (1) palpable taut band within the muscle; (2) presence of a hypersensitive spot in the taut band; (3) presence of a local twitch response of the taut band with palpation [26].

### 2.3. Instruments

#### 2.3.1. Tecar Therapy

In the first stage, sociodemographic and anthropometric data were collected. After collecting the data needed to perform the descriptive analysis, the intervention was conducted on the medial grastrocnemius with the T-CARE TECAR^®^ therapy machine (Florence, Italy) (see Figure 1). The therapy was applied with the generator emitting 0.5 MHz radiofrequency signals at a variable power with a maximum of 300 W. The frequency used was 500 MHz with an intensity of 40% (see Figure 2). The technique should be applied by direct contact with the body skin according to the protocol described in 2015 since it does not produce direct radiation [27,28]. The application protocol was implemented based on previous studies^1^. The intervention was performed by a physiotherapist with 5 years of experience in this technique.

The head of the device was applied over the treated area for 25 min (15 min of capacitive head and 10 min of resistive head), maintaining the same intensity and at all times. It was applied along and in circles over the medial gastrocnemius, generating light pressure above the muscle belly. The base plate was located below the region of the tibia to close the current circuit. The head of the device was applied over the entire medial gastrocnemius and not just over the latent MTP to improve the blood flow speed [29].

At baseline, data were collected through thermography (temperature), algometry (pain), and the lunge test (ankle ROM) before the intervention. All of the measurements were repeated immediately, 15′ after and 30′ after treatment. To eliminate any cross-effect that may interfere with the results, immediately after the post-treatment measurements, the participants were seated by research staff in a chair with 90° hip and knee flexion in a controlled room at a preset temperature (see Figure 3).

#### 2.3.2. Control Group

The same protocol as the Diathermy Group was performed but with the device in off mode (SHAM).

### 2.4. Procedures

#### 2.4.1. Thermographic Assessment

The thermography protocols were performed according to the International Academy of Clinical Thermology [27]. The equipment used were (Flir E6, FLIR Systems, Inc., Wilsonville, OR, USA) a step platform for the subject position and a black background to isolate body temperature.

For the thermographic assessment, the ideal temperature values (18–25 °C), relative humidity (39.8%), and atmospheric pressure (968 hPa were measured with the Ymiko Wifi digital weather station using a FLIR E60 camera according to Ring and Ammer (2000) [27,30,31] (see Figure 4).

The participants were placed barefoot on a 35-cm-high platform (Step) facing the wall with their hands on their waists with their bare legs 40 cm apart against a neutral black background so as to not interfere with body temperatures [32].

Images of both triceps surals were taken and interpreted with FLIR Tools software. Through the analysis, maximum (TMAX), minimum (TMIN), and mean (TMED) absolute temperatures were obtained from the medial gastrocnemius at the selected times.

#### 2.4.2. Algometry Assessment

A digital force gauge Algometer (model M3-20, 20 lbf, 10 KGF, 100 N) (Mark-10 Corporation, New York, NY, USA) was used by applying a pressure of 45 to 55 newtons on the MG and by also assessing the patient’s perceived pain using a VAS scale [13,33]. The Algometer was set to 0.09 kg/cm^2^ [34]. A 10-point VAS scale was used. The participants were asked to rate their pain by choosing a value (rating) on the 10-point VAS scale. The minimum value on the VAS was (0; no pain), meanwhile the maximum value was (10; maximum pain imaginable). A total of three measurements were taken, and the average value was taken into account for the analysis [3].

#### 2.4.3. Lunge Test Protocol

First, ankle ROM evaluation was performed using the Leg Motion system (Check Your Motion ^®^, Albacete, Spain) based on the ankle dorsiflexion test. This test evaluated the active ankle dorsiflexion in a standing position. The final test score was the distance between the first metatarsus and the wall when the participants bent their knee towards the vertical position without lifting the heel. The final range was measured in cm [35]. This test has been shown to have good reliability (ICC = 0.93–0.99) for ankle ROM in adults [36].

The reference values of ankle ROM restriction using Leg Motion^®^ has been established to be <11.5 cm and/or with a difference of 1.5 cm between the ankle ROM of both limbs. After the first measurement (baseline), a difference of 1.5 cm between limbs was detected for the ankle ROM restriction test. To become familiar with the test, the participants performed 3–4 attempts following the therapist’s instructions: (a) do not the heel from the surface; (b) try to take the knee as far as possible into the midline; (c) of not compensate the movement with the arms. The highest value of the 3–4 attempts was used for the analysis [36].

### 2.5. Statistical Analyses

All of variables were expressed as mean (M) and standard deviation (SD). For the calculation of the sample size, pain (assessed on VAS) was used as the main dependent variable. An alpha level of 0.05 and a desired power (beta) of 80% were used with the minimum value of the difference in VAS of 2.00, which is the minimum change needed to be considered clinically relevant [37]. The calculation generated a sample size of at least 15 participants per group after excluding 15% due to sample losses (3 patients). The sample size was consistent with a previous intervention study using tecar therapy [1]. To obtain the required sample size N, G*Power software Version 3 was used. All variables met the assumption of normality (i.e., Kolmogorov–Smirnov). In order to analyze the effect of diathermy on the mobility of the ankle joint, a repeated measurement analysis of covariance (ANCOVA) was performed, using the pre-intervention measurement as the covariate. On the other hand, a mixed factorial ANOVA was performed to analyze the effect of diathermy on muscle temperature. If the expected sphericity was not satisfied, the degrees of freedom were corrected according to the Greenhouse–Geisser correction. Bonferroni’s post hoc was used to analyze multiple comparisons. On the other hand, the effect size was expressed as the typified mean change difference, where the formula denominator used the pre-combined deviation of the two groups. The significance level was set at *p* < 0.05. All of the analyses were performed using statistical analysis software (SPSS Inc., Chicago, IL, USA) (SPSS Statistics 24.0 Mac version).

## 3. Results

Considering the pain values (i.e., hyperalgesia), the of the mixed factorial ANOVA showed statistically significant differences in the main effect of Time (F_[3.90]_ = 9.64, *p* = 0.001, eta = 0.24), showing no significant differences in the comparison Group x Time (F_[3.90]_ = 0.73, *p* = 0.539, eta = 0.02). In the within-group comparison, the Diathermy Group showed differences in the comparison of post-immediate vs. pre- (MD = 1.75, *p* = 0.003) and post-15 vs. pretreatment (mean difference; MD = 1.56, *p* = 0.005). In relation to the Control Group, Bonferroni’s post hoc showed statistically significant differences in the comparison of post immediate vs. pre-captures (MD = 1.29, *p* = 0.046). No statistically significant differences were found between any of the possible between-group comparisons (*p* > 0.05).

In regard to ankle mobility, RM ANCOVA showed no statistically significant differences (F_[1.68, 53.64]_ = 0.02, *p* = 0.980, η^2^_p_ = 0.001) in the main effect of time (i.e., post immediate, post 15 min and post 30 min). After transforming the scores of the time variable regarding to pre-intervention values, the model was corrected to F_[1.68, 53.64]_ = 2.91, *p* = 0.072, η^2^_p_ = 0.08, without ever obtaining statistically significant differences. Figure 5 summarizes the MD for each of the conditions.

The Bonferroni post hoc analysis showed that there were no statistically significant differences in the comparison of the variables between the groups (e.g., Experimental Group = 9.36 cm vs. Control Group: CG = 9.10 cm, *p* = 0.213). However, statistically significant differences were found in the comparison of the main effect of the time variable (MD Immediate vs. 30 min (MD_immvs.30m_ = −0.413 cm, *p* = 0.001)) and post-15 min vs. 30 min (MD_post15vs.post30_ = −0.236 cm, *p*= 0.040).

On the other hand, no statistically significant differences were found in the effect of *Group x Time* interaction (F[2,64] = 0.16, *p* = 0.814, η^2^_p_ = 0.005). Bonferroni’s post hoc analysis showed no statistically significant differences for any of the comparisons (i.e., post-immediate measurement (CG = 8.93 cm vs. EG = 9.13 cm, *p* = 0.348), measurements post-15 min (CG = 9.08 CM vs. EG = 9.34 cm, *p* = 0.202), and measurements post-30 min (CG = 9.29 CM vs. EG = 9.59 cm, *p* = 0.263)). Figure 6 shows the effect size for each of the comparisons between the Control Group vs. the Experimental Group.

Regarding muscle temperature, the mixed factorial ANOVA showed significant differences in the main effect for *Group* (F_[1,62]_ = 4.60, *p* = 0.040, η^2^_p_ = 0.13) and in the interaction between *Group x Capture* (F_[3,186]_ = 7.08, *p* = 0.001, η^2^_p_ = 0.19). Bonferroni’s post hoc analysis showed statistically significant differences between the Diathermy Group vs. the Control Group (MD = 0.44 °C, *p* = 0.040, _95%_ CI = 0.02, 0.85). Figure 7 summarizes the within- and between-group differences. Figure 8 shows different pre-and post-intervention captures of a subject.

## 4. Discussion

The main objective of the study was to analyze the acute effect of TT on latent MTPs on skin temperature, ankle ROM, and pain in professional male basketball players. Our results showed a significant increase in the absolute temperature in the diathermy group, with no significant differences within and between groups in the rest of the parameters. An increase in pain (VAS) was shown in the TT group, which describes an increased sensitivity in the MTP following the application of this technique. Currently, only few studies have analyzed the effects of TT applied on medial gastrocnemius MTPs in basketball players on pain. However, to our knowledge, this is the first study to evaluate the effects of TT on temperature, hyperalgesia, and ankle ROM.

TT produces a thermal effect, assumed to be a biological effect related to hyperthermia. Our results are similar to previous studies regarding the temperature due to its increase after applying the technique [4]. For instance, the study of Benito et al. [17] found no significant differences between groups in skin temperature assessed by thermography, as was also the case in our study. The interaction of radiofrequency currents with biological structures results in an increase in endogenous temperature [38]. In our study in particular, changes in the neuromuscular region corresponding to latent MTPs were evidenced. Previously, it has been stablished that high-intensity treatments increased the temperature and significantly raised blood vascularization in the Achilles tendon [24].

By contrast, other investigations found no significant changes when TT was applied over other anatomical regions. For instance, TT did not alter the circulation in the peritendinous region of the Achilles tendon [5].

Furthermore, the present study noted that after 15 min of the application, there is a 1.5° drop from the absolute temperature peak. According to Kumara and Watson’s study on the thermophysiological effects of the skin after applying TT, this technique was compared to pulsed short-wave therapy, which obtained better results than those in the TT group [39]. These findings are consistent with those obtained in our work, where after applying TT, the temperature immediately rose by 1.5 points to generate changes and improvements in tissues, which were mediated by the impact of pyogenic precursor cells [3]. The increase in temperature immediately after the intervention has been studied and evaluated in connection with the clinical effects of brachioradial pain [40]. A significant difference was also generated 15 min after the intervention.

Regarding ankle mobility, we did not find any significant improvement, which was in contrast with previous investigations. MTPs may adversely affect clinical effects on restricted ROM [41]. According to Hong-You et al., restricted joint ROM is commonly observed in health people when latent MTPs [16] are present because they can produce a series of neuromuscular disorders, such as inefficient muscle contraction [42].

Current evidence describes an increase in tissue elasticity after TT [4]. Such increased tissue flexibility was not evident in ankle ROM measurements in the current study. Undoubtedly, the importance of the presence of latent MTPs as a potential dysfunction must be highlighted [13]. In contrast, one study established a positive association between skin temperature and MPS-related ROM [43]. Accordingly, the evaluation of dorsal flexion in our study was justified by the lunge test.

Various treatments for MTPs have been shown to improve clinical outcomes, including improved muscle strength, ROM, and pain reduction [44]. For example, ROM improved after the application of other therapies used to treat MTP, such as dry needling [45]. Grieve et al. achieved a 5-degree improvement in ankle dorsiflexion after treating active and latent MTPs in the soleus muscle [46].

Applying TT alone on a MG latent MTP does not produce changes in ROM when it is measured with the lunge test. On the other hand, Draper et al. reported that the combination of TT and joint mobilization is considered to be an effective treatment to recover joint ROM in elbow extension in post-surgical patients [47].

Despite our results, other studies that have used the same treatment technique have found an increase in ROM in other joints, such as in the right rotation of the cervical spine [3]. However, it should be noted that these were chronic effects, not acute as they were in our study, since they were obtained after eight sessions.

Regarding pain, we did not find any changes in pain assessed by VAS scale. This result is in line with the study of Aguacil I et al. [3], which did not find significant differences between the Control Group and the SHAM group after TT [3]. However, other authors [1] have shown that TT is able to accelerate the fatigue recovery in runners [1]. Moreover, Paulocci et al. [4] described that TT can reduce pain and can improve quality of life [4].

The assessment of the sensitivity of the palpable nodule in the latent MTP through algometry justifies the VAS measurement evaluation of this study. An advantage of TT was that pain scarcely increased compared to other techniques commonly used in physiotherapy, although an increase in hyperalgesia was noted in the post-immediately measurement.

Some possible factors that can explain the reduction of hyperalgesia on VAS thirty minutes after treatment that we can find are the natural evolution of the disease [48]; the specific perceived pain of each subject and how this develops over time [48,49]; and the active physiological mechanisms originating from psychological processes (placebo effect) [49]. These factors can also explain why there were small differences between the two groups. However, a 10% decrease in mechanical hyperalgesia was observed when stimulated by treatment 30 min after the intervention (P30).

Contrary to our results, other similar treatments were effective in reducing mechanical hyperalgesia compared to control groups. The main difference was that the technique was applied on active MTPs rather than on latent MTPs [50]. Despite this clinical difference, an increase in generalized hyperalgesia pressure can be found in asymptomatic participants with latent MTPs [51], and their treatment may reduce latent MTP hyperalgesia.

Furthermore, previous studies have shown that pressure inhibition treatment reduced pain in MTPs in the upper trapezius [42]. These studies are not in line with our results since hyperalgesia was observed immediately after the intervention but was normalized after 30 min. Thus, TT produces immediate but momentary hyperalgesia.

If we extrapolate the effects of TT in patients diagnosed with different pathologies, the literature shows that TT is a useful therapy for reducing pain in patients with osteoarthritis [9]. In other words, pain reduction is one of the main effects of TT, although more research is needed regarding other physiological aspects. The increase in temperature reduces pain by promoting the vasodilation of tissues affected by pain mediators, such as bradykinin, serotonin, and prostaglandin [8].

Although no promising changes in the studied variables were obtained, it has been observed that currently, there are a wide range of standardized protocols for the application of this technique [52]. This means that it is important to highlight the difficulty of establishing the most specific protocol [8]. Although the effect of TT on latent MTPs assessed through the variables analyzed in the present study was small, it is possible that the evaluation of other parameters or the application of other parameters (protocol) could determine the effectiveness of TT on myofascial pain. New lines of research should be explored in relation to the development of the technique associated with specific pathologies. It is necessary and highly important to agree on a standardized TT protocol according to specific pathologies.

The theoretical consequences of this work entail the need to evaluate this technique using other methods or assessment tests and to verify or contrast our results. The importance of this lies in the possibility of a new hypothesis and new experimental studies. Possible practical applications of this study are related to the treatment of muscle stiffness in athletes in a non-invasive, painless, and effective way of using TT.

The present study is not without limitations. Although thermography is an innovate technique for measuring skin temperature, we extrapolate that the external temperature reflects the internal or muscle temperature. In addition, the lack of reliability analysis of the algometer assessment could be also considered to be a limitation.

Future research with larger sample sizes should examine further does–response relationships between TT and other physiological parameters in latent and active MTPs. In addition, further studies are needed to compare the effects of TT against other treatment modalities.

## 5. Conclusions

In conclusion and according to our results, TT can induce temperature changes in the medial gastrocnemius in professional male basketball players, generating an increase of local temperature and a decrease of local pain (VAS) at the MTP after treatment. In addition, TT does not affect the ankle dorsiflexion ROM.

These results suggest that use of TT could be useful to improve muscle recovery. Further studies with similar characteristics and implemented during an experimental period to assess chronic adaptations would be needed to gain in-depth knowledge of the advantages of this technique.

## Figures and Tables

**Figure 1 ijerph-18-08756-f001:**
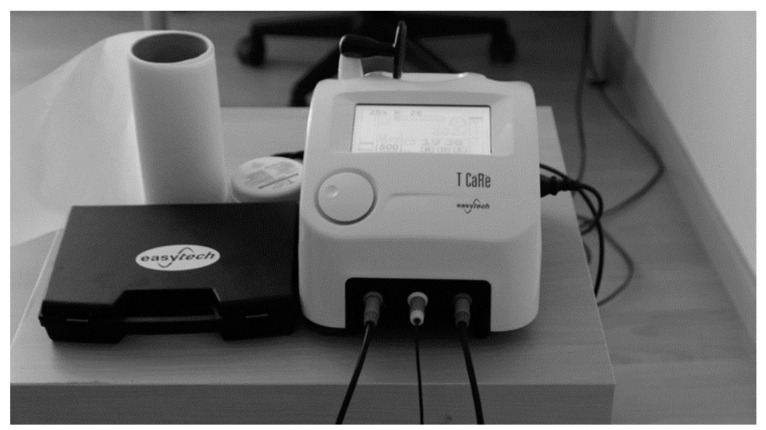
T-Care TECAR^®^ therapy device. Prepared by the authors.

**Figure 2 ijerph-18-08756-f002:**
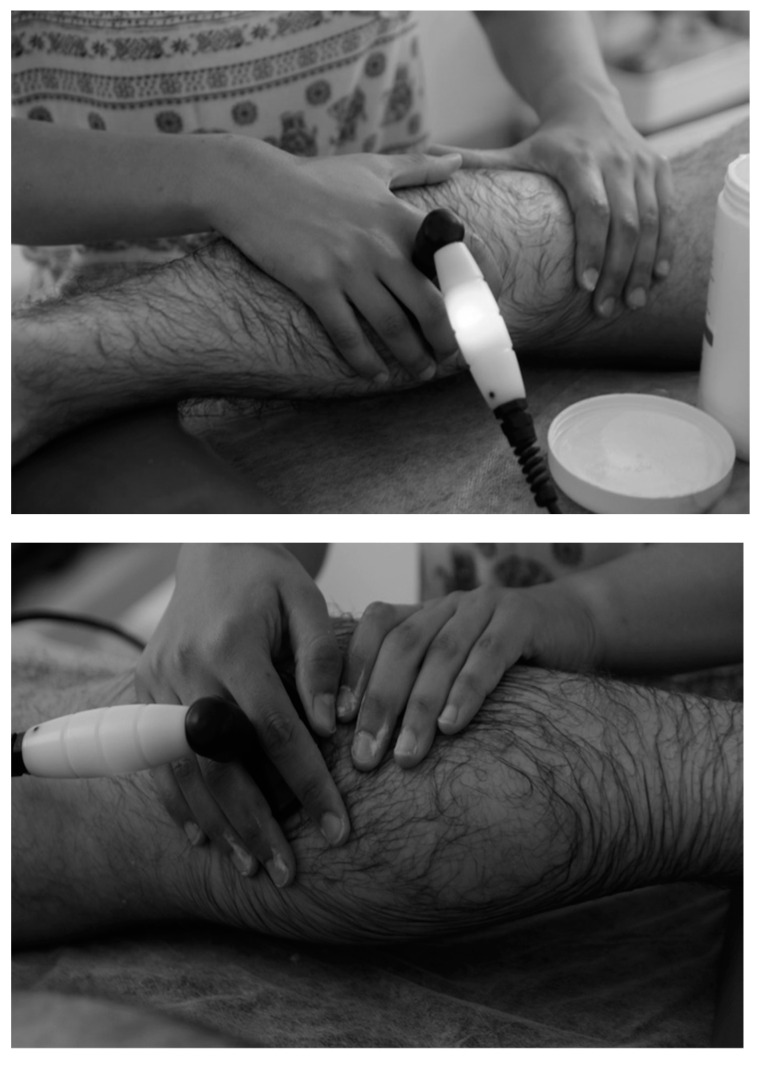
Patient in comfortable prone position, adhesive electrode on anterior tibial area (10 × 15 cm). Therapist’s hand holding the capacitive head. Therapist’s hand holding the resistive head. Prepared by the authors.

**Figure 3 ijerph-18-08756-f003:**
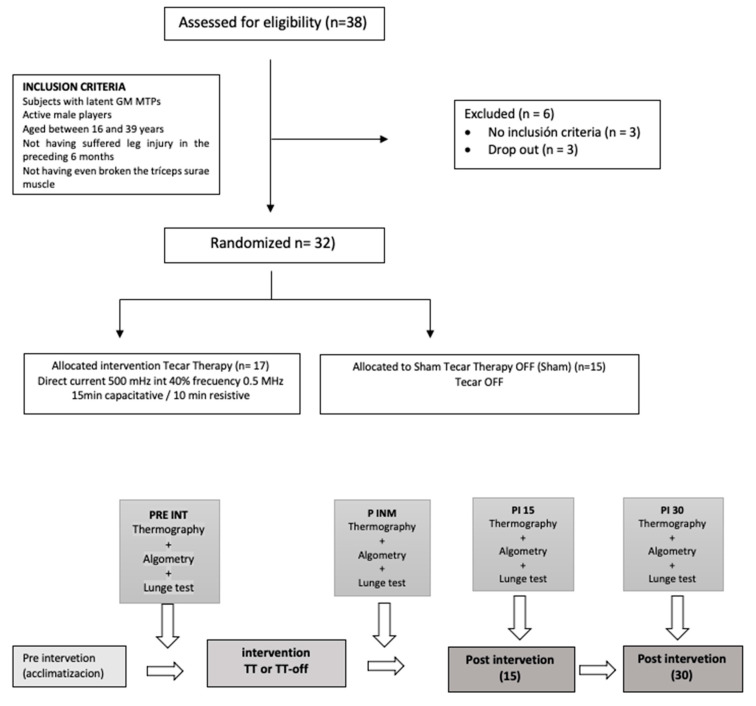
Flow diagram of the intervention procedure. Intervention process of the diathermy group and the Control Group. Prepared by the authors. (Latent GM MTP’s: Latent medial gastrocnemius myofasical trigger point; PRE INT: pre intervention; P INM: post immediately; P15: after 15 min intervention; PI 30: after 30 min intervention).

**Figure 4 ijerph-18-08756-f004:**
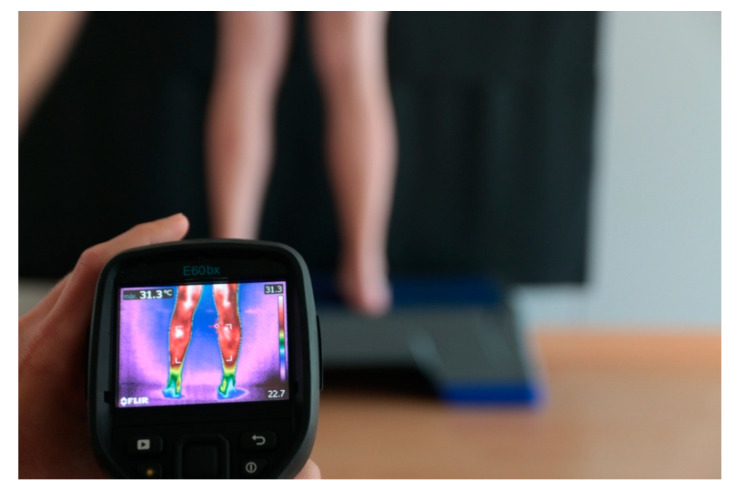
Image capture with FLIR E60. Prepared by the authors.

**Figure 5 ijerph-18-08756-f005:**
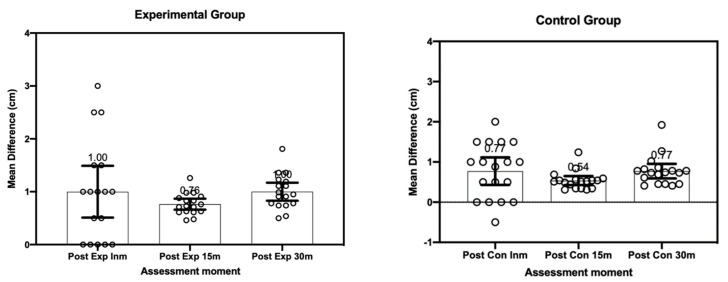
Differences in ankle mobility (in cm) in each of the intervention conditions. The figure shows the values of the difference between the conditions obtained by each subject (circles) both in the experimental conditions and in the control conditions. Bars represent the mean value, while boxplots represent the 95% confidence interval. The mean value of each condition is represented at the top edge of the CI. Prepared by the authors. (Exp: experimental group; Con: Control; Post INM: post immediately; P15: after 15 min intervention; PI 30: after 30 min intervention).

**Figure 6 ijerph-18-08756-f006:**
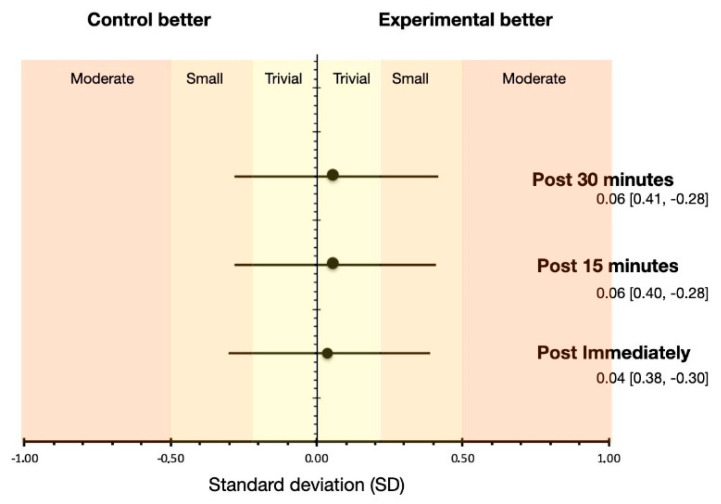
Effect size (ES) in each of the comparisons of the time variable (i.e., post immediate, post-15 min, and post-30 min). Prepared by the authors. (Post: post intervention).

**Figure 7 ijerph-18-08756-f007:**
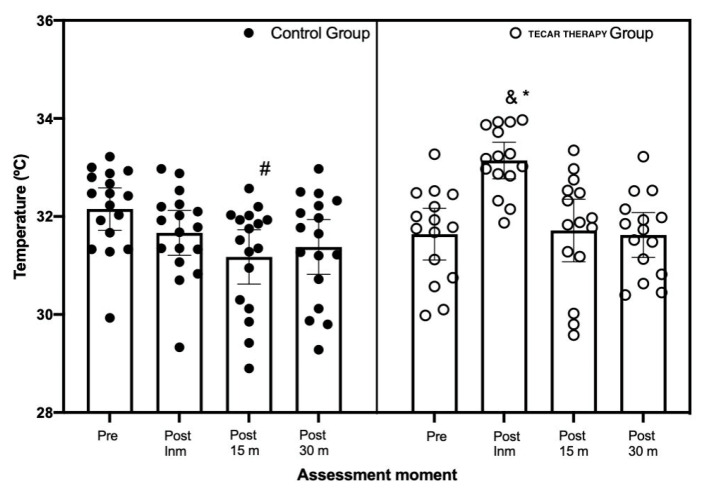
Mean, minimum, and maximum values of muscle temperature (°C) in the two intervention groups in relation to the measurements taken (i.e., Pre, post-immediate, post-15 min, and post-30 min). Note: * significant differences (*p* < 0.05) compared to the pre series in the tecar therapy group. # Number of significant differences (*p* < 0.05) compared to the pre series in the Control Group, & and significant differences (*p* < 0.05) in the Control Group and tecar therapy group comparison. The lines represent the mean value and 95% confidence intervals. Prepared by the authors. (Pre: pre intervention; Post INM: post immediately; P15: after 15 min intervention; PI 30: after 30 min intervention).

**Figure 8 ijerph-18-08756-f008:**
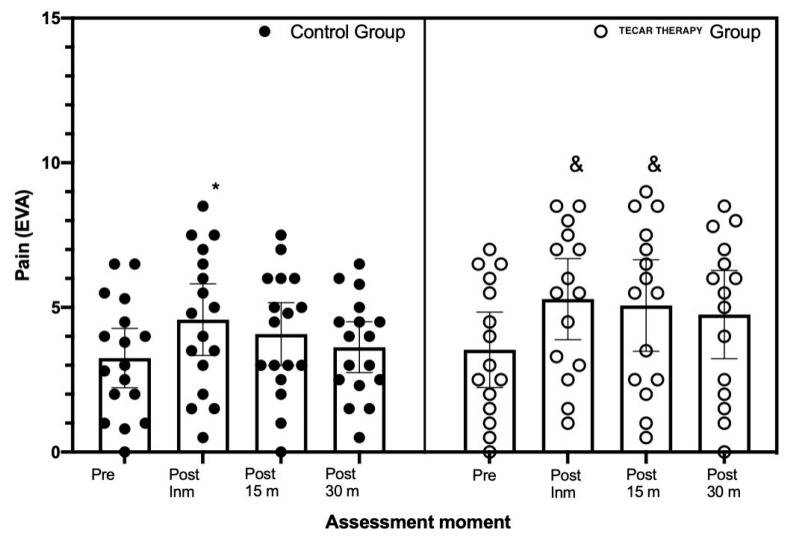
Mean, minimum, and maximum values of hyperalgesia (VAS scale) in the two intervention groups in relation to the measurements taken (i.e., pre, post-immediate, post-15 min, and post-30 min). Note: * significant differences (*p* < 0.05) compared to the pre series in the Control Group, and & significant differences (*p* < 0.05) compared to the pre series in the tecar therapy group. The lines represent the mean value and 95% confidence intervals. Prepared by the authors. (Pre: pre intervention; Post INM: post immediately; P15: after 15 min intervention; PI 30: after 30 min intervention).

## Data Availability

Data sharing is not applicable to this article.

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
