# Peer review of "Acute Effects of Tecar Therapy on Skin Temperature, Ankle Mobility and Hyperalgesia in Myofascial Pain Syndrome in Professional Basketball Players: A Pilot Study"

_ijerph, 2021, doi:10.3390/ijerph18168756_

Round 1

Reviewer 1 Report

Thank you for asking me to review this paper on an RCT of the effects of electro-thermal therapy on ankle range of motion, skin temperature and pain response in basketball players. I have a few questions and suggestions for the authors.

General comments

The paper lacked clarity on several methodological points. Consider using the CONSORT statement to ensure you include all pertinent information. https://www.equator-network.org/reporting-guidelines/consort/

Sections of the paper were difficult to follow. At times I was unsure if it was the logical flow or the use of English that made it difficult. Consider fully revising your paper with a native English speaker so there is clarity of expression.

Take care with your referencing. At several points you reference a paper in support of your argument, but on examination the referenced paper does not support the argument but has within it a reference to a paper that would support your argument. You should always reference the original source. I have put examples in the introduction section below but have not checked all references in your paper.

The title and the aim appear at odds: the title says the paper investigated TT in basketball players with myofascial pain syndrome, but the aim and the inclusion criteria say it was myofascial trigger points?

Many abbreviations ended up being used in the paper that are not standard abbreviations. I know this helps with the word limit, but it makes reading the article very difficult. Consider reducing the number of abbreviations.  E.g. MG, LT, TM, AL, ST, RF,

Specific comments

Abstract

Unclear what ‘This technique is characterised by its speed of action being used by high performance athletes’ means. Please clarify.

Introduction

The logical flow of the introduction was difficult to follow leading to a poor justification for the study. Just because something has not been investigated doesn’t mean it should be.

Line 38 The first sentence contains ‘also’ but there has been no previous statement? Take care as the first few sentences are very similar to sentences found in the Dunabeitia article.

Line 41 reference 1 does not support your argument here- use the original reference cited in the reference paper.

Line 41 Reference 2 does not support your argument that RF energy is the most commonly used energy source to generate therapeutic heat or in muscle stiffness. Is this true in all parts of the world? Other clinicians may treat muscle stiffness another way, or use heat from heat packs etc. Rephrase or re-reference.

Line 46 ‘compressed generate movement that produces painful vegetative alterations’ does not make sense. Please clarify

Line 47 Is MPS one of the most common causes of muscle pain? Please reference.

Line 48 Reference 4 is not the correct reference. The correct reference is Tough EA, White AR, Richards S, et al. Variability of criteria used to diagnose myofascial trigger point pain syndrome—evidence from a review of the literature. Clin J Pain 2007;23:278–86.

Line 48 Take care to specifically say that 85% of patients at ONE pain clinic, not imply that this is true at all physiotherapy centres.

Line 51 I am not sure that latent MTPs are a key for injury prevention? How? What is the evidence for this? Your reference (6) is a systematic review of the effects of dry needling on musculoskeletal pain. Maybe there is a reference within the paper you are referring to?

Line 54 Reference the paper where Travell and Simons used the term.

Line 56 What stimulus response in induced by pressure?

Line 64 Do latent MTPs restrict ankle ROM? Ref 1 did not mention ankle ROM as an outcome measure, and I could not find reference 9 at the website mentioned.

Line 66ff There is a link missing from saying that sural injuries are commonly in the medial gastrocs, to talking about myofascial pain. What is the link?

Line 77 Your reference (17) does not support your statement that TT accelerates the recovery process. In fact there was no difference in pain between the groups.

Line 81 The Visconti study actually showed no difference between the groups, including a sham, and did not have an outcome that looked at MTPs, so how does this justify the statement that the treatment is effective in treating later MTPs?

Materials and Methods

Several methods are unclear. Please clarify the following:

  1. How did you diagnose latent MTP’s? Who did the diagnosis?
  2. Clarify your single blinding- was it the person who applied the treatment who was blinded? Was it the outcome assessor who was blinded?
  3. Does the thermographic equipment measure skin temperature only or does it imply temperature withing the muscle? How do you end up with a range of temperatures? If it is over an area then how did you standardise the area measured?
  4. Where was the algometer applied? Was any reliability of measurements done?
  5. Was VAS scale was used? 10 point or 100point? If it was a VAS what were the anchor points for your scale and how did you measure the line? If it was verbal then it was not a VAS but a numerical rating scale.
  6. Why was the 3rd or 4th attempt used for the lunge test? What determined which one was used?

Line 89 Somewhere there should be a reporting of the characteristics of the 2 groups, not just the overall cohort.

Line 177 This section on how you determined a difference between legs should go in the inclusion criteria section.

Line 202 which version of SPSS was used? If it was a version since IBM brought the software then you should reference it as IBM requests.

Results

Consider reordering your results so report pain first- since this was your main outcome and you calculated your sample size based on this.

For your results where the ANCOVA and ANOVA were not statistically significant, you should not perform a post hoc analysis since your primary analysis is not significant.

Line 218: What is the Capture variable? It has not been mentioned previously and should be included in the methods section.

Discussion

Line 263 You set your significance level at 0.05 so you should not talk about a positive trend. It was not significant.

Line 266 Can you say the effects of TT as a recovery technique when you did not measure recovery? Do you mean on sensitivity of the MTP?

Line 269 ff when referencing other studies, be clear when you are referring to your ‘present’ study and when you are referring to ‘other’ studies.

Line 280 ff At line 280 you say there was a drop of 1.5 deg in temperature peak but at line 286 you imply that your results are similar to a study where an increase in temperature occurred?

Line 290-295 Should this paragraph be in the ROM section?

Section 4.2 was difficult to follow. Suggest reworking the section to start with’ We did not find any changes to ankle ROM in contrast to other studies. Then discuss the other studies and why their result may be different to yours.

Section 4.3 was difficult to follow. Suggest starting with your result- no change in pain and then discussing studies with similar and different results.

Conclusions

Your conclusion is slightly misleading- you did find TT reduced pain, but so did your sham group. Can you recommend a treatment be applied, using time rand resources, that is no more effective than a sham treatment?

 Figures

Results figures should have any abbreviations described in the legend. E.g. Post Con Inm

Figures 7 and 8 Do you mean diathermy group?

Author Response

Reviewer response 1

REVIEWER: Thank you for asking me to review this paper on an RCT of the effects of electro-thermal therapy on ankle range of motion, skin temperature and pain response in basketball players. I have a few questions and suggestions for the authors.

AUTHORS: We would like to thank the reviewer for his/her valuable comments, which are really constructive and have definitely helped us to improve the quality of the present manuscript.

REVIEWER: The paper lacked clarity on several methodological points. Consider using the CONSORT statement to ensure you include all pertinent information. https://www.equator-network.org/reporting-guidelines/consort/

AUTHORS: We have reviewed the CONSORT statement to make sure that every point is included and, in general, all the pertinent information.

REVIEWER: Sections of the paper were difficult to follow. At times I was unsure if it was the logical flow or the use of English that made it difficult. Consider fully revising your paper with a native English speaker so there is clarity of expression.

AUTHORS: Our manuscript has been reviewed by a professional native speaker to improve the readability.

REVIEWER: Take care with your referencing. At several points you reference a paper in support of your argument, but on examination the referenced paper does not support the argument but has within it a reference to a paper that would support your argument. You should always reference the original source. I have put examples in the introduction section below but have not checked all references in your paper.

AUTHORS: We have checked the references of the paper.

REVIEWER: The title and the aim appear at odds: the title says the paper investigated TT in basketball players with myofascial pain syndrome, but the aim and the inclusion criteria say it was myofascial trigger points?

AUTHORS: Sensitive areas of tight muscle fibers can form in your muscles after injuries or overuse. These sensitive areas are called trigger points. A trigger point in a muscle can cause strain and pain throughout the muscle. When this pain persists and worsens, produce myofascial pain syndrome.

REVIEWER: Many abbreviations ended up being used in the paper that are not standard abbreviations. I know this helps with the word limit, but it makes reading the article very difficult. Consider reducing the number of abbreviations.  E.g. MG, LT, TM, AL, ST, RF,

AUTHORS: We have reduced the number of abbreviations, as you suggested.

REVIEWER: Unclear what ‘This technique is characterised by its speed of action being used by high performance athletes’ means. Please clarify.

AUTHORS: We have refreshed the sentence to clarify (in page 2, line 29)

REVIEWER: Line 38 The first sentence contains ‘also’ but there has been no previous statement? Take care as the first few sentences are very similar to sentences found in the Dunabeitia article.

AUTHORS: We have changed it. (Page 2, line 47)

REVIEWER: Line 41 reference 1 does not support your argument here- use the original reference cited in the reference paper.

AUTHORS: We have added one more reference in this part of the paper (Page 2, line 50).

REVIEWER: Line 41 Reference 2 does not support your argument that RF energy is the most commonly used energy source to generate therapeutic heat or in muscle stiffness. Is this true in all parts of the world? Other clinicians may treat muscle stiffness another way, or use heat from heat packs etc. Rephrase or re-reference.

AUTHORS: We have refreshed the sentence to clarify (page 2, line 50-52).

REVIEWER: Line 46 ‘compressed generate movement that produces painful vegetative alterations’ does not make sense. Please clarify

AUTHORS: We have changed this part (page 3, line 63-66)

REVIEWER: Line 47 Is MPS one of the most common causes of muscle pain? Please reference.

AUTHORS: We have added a reference (Page 3, Line 65-66).

REVIEWER: Line 48 Reference 4 is not the correct reference. The correct reference is Tough EA, White AR, Richards S, et al. Variability of criteria used to diagnose myofascial trigger point pain syndrome—evidence from a review of the literature. Clin J Pain 2007;23:278–86.

AUTHORS: We have changed the reference (Page 3, Line 63-65).

REVIEWER: Line 48 Take care to specifically say that 85% of patients at ONE pain clinic, not imply that this is true at all physiotherapy centres.

AUTHORS: We have refreshed this part (Page 3, Line 66) .

REVIEWER: Line 51 I am not sure that latent MTPs are a key for injury prevention? How? What is the evidence for this? Your reference (6) is a systematic review of the effects of dry needling on musculoskeletal pain. Maybe there is a reference within the paper you are referring to?

AUTHORS: We have changed this part (Page 3, Line 71-72) .

REVIEWER: Line 54 Reference the paper where Travell and Simons used the term.

AUTHORS: We have added the reference (Page 3, Line 74).

REVIEWER: Line 56 What stimulus response in induced by pressure?

AUTHORS: We have added a reference and changed the sentence (Page 3, Line 74-80).

REVIEWER: Line 64 Do latent MTPs restrict ankle ROM? Ref 1 did not mention ankle ROM as an outcome measure, and I could not find reference 9 at the website mentioned.

AUTHORS: We have changed the reference and refreshed the reference (Page 3, Line 84-85).

REVIEWER: Line 66ff There is a link missing from saying that sural injuries are commonly in the medial gastrocs, to talking about myofascial pain. What is the link?

AUTHORS: Sorry for the mistake, we have changed the sentence (Page 3, Line 87-93).

REVIEWER: Line 77 Your reference (17) does not support your statement that TT accelerates the recovery process. In fact there was no difference in pain between the groups.

AUTHORS: We have changed it for the correct reference, as you suggested (Page 2, Line 58-60).

REVIEWER: The Visconti study actually showed no difference between the groups, including a sham, and did not have an outcome that looked at MTPs, so how does this justify the statement that the treatment is effective in treating later MTPs?

AUTHORS: We apologize for this mistake, and we have corrected the text, as suggested.

REVIEWER: How did you diagnose latent MTP’s? Who did the diagnosis?

AUTHORS: Latent TrPs were identified as follows: 1) palpable taut band within the muscle; 2) presence of a hypersensitive spot in the taut band; and 3) presence of a local twitch response of the taut band with palpation1.

REVIEWER: Clarify your single blinding- was it the person who applied the treatment who was blinded? Was it the outcome assessor who was blinded?

AUTHORS: The person who applied the treatment was not blinded. However, the outcome assessor was blinded, he did not if the participants were in the control group or not.

REVIEWER: Does the thermographic equipment measure skin temperature only or does it imply temperature withing the muscle? How do you end up with a range of temperatures? If it is over an area then how did you standardise the area measured?

AUTHORS: We have now marked the point we wanted to measure and then we captured the thermographic picture perpendicular to it. The camera software determines through the pixels the average of minimum and maximum temperatures, achieving averages of the area as you delimited.

We use the thermography as showing that skin temperature within muscle temperature. We understand the limitation of thermography.

REVIEWER: Where was the algometer applied? Was any reliability of measurements done?

AUTHORS: Yes, we are aware that, in the first, the reliability is not the best practice but at this point we cannot do anything else. In this sense, we have rewrited the limitation. The algometer was applied in myofascial trigger point and we can not perform any reliability measurement but reliability of any study shows values (95% IC) for rather 1 and 2 were 0.97 (0.95-0.98) and 0.84 (0.73-0.90) respectively2

REVIEWER: Was VAS scale was used? 10 point or 100point? If it was a VAS what were the anchor points for your scale and how did you measure the line? If it was verbal then it was not a VAS but a numerical rating scale.

AUTHORS: We have used 10-point VAS scale, the subjects were asked to rate a pain, choosing a value (rating) on the 10-point VAS scale. The minimum value on the VAS scale was 0 whereas the maximum value scale was 10. We have added this information in the paper (Page 6, Line 187-190).

REVIEWER: Why was the 3rd or 4th attempt used for the lunge test? What determined which one was used?

AUTHORS: We have used the highest values of 3 attempts based on of the previous literature3.

REVIEWER: Line 89 Somewhere there should be a reporting of the characteristics of the 2 groups, not just the overall cohort.

AUTHORS: This sentence was a mistake. We deleted.

REVIEWER: Line 177 This section on how you determined a difference between legs should go in the inclusion criteria section.

AUTHORS: We have refreshed the sentence (Page 4, Line 126-131).

REVIEWER: Line 202 which version of SPSS was used? If it was a version since IBM brought the software then you should reference it as IBM requests.

AUTHORS: We used the software SPSS Statistics 24.0, Mac version (Page 7, Line 228).

REVIEWER: Consider reordering your results so report pain first- since this was your main outcome and you calculated your sample size based on this.

AUTHORS: We have refreshed the sentence.

REVIEWER: For your results where the ANCOVA and ANOVA were not statistically significant, you should not perform a post hoc analysis since your primary analysis is not significant.

AUTHORS: After correcting the variance, having used a covaried model, we found a snedecor f of 2.91, getting a p value of 0.072. Despite of non-statistically significative effects, were found that the covariated model increased.

REVIEWER: Line 218: What is the Capture variable? It has not been mentioned previously and should be included in the methods section.

AUTHORS: We have changed capture for time, we mean measures before and after treatment (before treatment, after treatment immediately, after 15 minutes and 30)

REVIEWER: Line 263 You set your significance level at 0.05 so you should not talk about a positive trend. It was not significant. 

AUTHORS: We have changed the sentence (page 12)

REVIEWER: Line 266 Can you say the effects of TT as a recovery technique when you did not measure recovery? Do you mean on sensitivity of the MTP?

AUTHORS: We have refreshed the sentence (Page 12) .

REVIEWER: Line 269 ff when referencing other studies, be clear when you are referring to your ‘present’ study and when you are referring to ‘other’ studies.

AUTHORS: We have changed the sentence (Page 12, line 411) .

REVIEWER: Line 280 ff At line 280 you say there was a drop of 1.5 deg in temperature peak but at line 286 you imply that your results are similar to a study where an increase in temperature occurred?

AUTHORS: Yes, we are doing a comparison between our study and others’ where they experimented an increase of skin temperature after tecar therapy application.

REVIEWER: Line 290-295 Should this paragraph be in the ROM section?

AUTHORS: We have changed it (Page 12, Line from 442) .

REVIEWER: Section 4.2 was difficult to follow. Suggest reworking the section to start with’ We did not find any changes to ankle ROM in contrast to other studies. Then discuss the other studies and why their result may be different to yours.

AUTHORS: We have refreshed it.

REVIEWER: Section 4.3 was difficult to follow. Suggest starting with your result- no change in pain and then discussing studies with similar and different results.

AUTHORS: We have changed it.

REVIEWER: Your conclusion is slightly misleading- you did find TT reduced pain, but so did your sham group. Can you recommend a treatment be applied, using time rand resources, that is no more effective than a sham treatment?

AUTHORS: In our study, the applied treatment did not show any pain changes, existing different techniques: invasive and non-invasive. In one hand, there are invasive techniques such as dry needling, percutaneous needle electrolysis, neuromodulation or infiltrations. On the other hand, there are non-invasive techniques such as stretching, massage therapy or foam roller ones.

REVIEWER: Results figures should have any abbreviations described in the legend. E.g. Post Con Inm

AUTHORS: We have described the legends of all the figures in the paper in order to make it clearer.

REVIEWER: Figures 7 and 8 Do you mean diathermy group?

AUTHORS: Diathermy group mean tecar therapy group, diathermy and tecar have the same meanings, but thank you for the clarification.

References:

1         Fernández-de-las-Peñas C, Dommerholt J. International consensus on diagnostic criteria and clinical considerations of myofascial trigger points: A delphi study. Pain Medicine (United States) 2018; 19: 142–50.

2         Pelfort X, Torres-Claramunt R, Sánchez-Soler JF, et al. Pressure algometry is a useful tool to quantify pain in the medial part of the knee: An intra- and inter-reliability study in healthy subjects. Orthopaedics and Traumatology: Surgery and Research 2015; 101: 559–63.

3         Cejudo A, Sainz de Baranda P, Ayala F, Santonja F. A simplified version of the weight-bearing ankle lunge test: Description and test-retest reliability. Manual Therapy 2014; 19: 355–9.

Reviewer 2 Report

This study is very interesting. However, a few more modifications need to be made. Please revise it after checking the comment.

- Please check again whether the described abstract is the form of IJERPH.

- What's TECAR?, Product name? Describe the full name.

- The author presented Diathermy in the title. However, the main word presented in the study is TECAR Therapy. I think it's good that the author emphasizes Diathermy. So I hope that the research will unify some of the terms.

- Tecar?, TECAR?, Match it in the text.

- Isn't there already a lot of research on Diathermy? It's judged that there's a lot about electric-based treatments. What does the author think is unique about this method compared to conventional electric-based treatments? There is a lack of explanation for how this study differs from existing studies. It should be possible to demonstrate the novel of this work while presenting previous studies.

- Create 2.1 Subjects in Materials and Methods.

- Basketball players' career, and basketball positions? Add this information.

- Modify 2.2 Tecar therapy group to Tecar therapy.

- Figure 3 is not clearly visible. It would be good to expand Figure further. It's hard to see what's inside the box.

- Describe the Control group more specifically.

- Describe the information about the thermographic assessment tool and Algometer (model M3-20, 20lbf, 10 KGF, 100N), and Leg Motion system (Check 170 Your Motion ® , Albacete, Spain). For example, (Model, Company, City, Nation). Match this part in this manuscript.

- Describe the discussion in an integrated way, not in each title. Interpret the results of this study by deleting unnecessary content and presenting only necessary parts. Describe the discussion by comparing the results of previous studies.

Author Response

Reviewer response 2

REVIEWER: This study is very interesting. However, a few more modifications need to be made. Please revise it after checking the comment.

AUTHORS: First of all, thank you very much for reviewing this article. We greatly appreciate the effort that this implies, having helped us to improve this paper.

REVIEWER: Please check again whether the described abstract is the form of IJERPH.

AUTHORS: We have checked the structure of the abstract in order to make sure that it follows the guidelines indicated by IJERPH; additionally, we have also added some numerical results to complement it.

REVIEWER: What's TECAR?, Product name? Describe the full name.

AUTHORS: TECAR means Transfer Energetic Capacitive and Resistive: Capacitive and resistive energetic transfer. Tecar therapy is an endogenous thermotherapy used to generate warming up of superficial and deep tissues.

REVIEWER: The author presented Diathermy in the title. However, the main word presented in the study is TECAR Therapy. I think it's good that the author emphasizes Diathermy. So I hope that the research will unify some of the terms.

AUTHORS: Diathermy and Tecar therapy means the same, but we thought it would be a good idea to change this word on the title.

REVIEWER: Tecar?, TECAR?, Match it in the text.

AUTHORS: We have checked and changed it.

REVIEWER: Isn't there already a lot of research on Diathermy? It's judged that there's a lot about electric-based treatments. What does the author think is unique about this method compared to conventional electric-based treatments? There is a lack of explanation for how this study differs from existing studies. It should be possible to demonstrate the novel of this work while presenting previous studies.

AUTHORS: Diathermy works with alternating currents higher than 400 Khz, which generates a heating without risk. There are other types of thermotherapy, as infrared thermotherapy that generates an increase of temperature but the heat does not penetrate, it fundamentally warms the skin. Instead, diathermy can produce a deeper increase of temperature.

The innovate technical characteristics of tecar therapy allow the transfer of high energy levels without causing significant temperature increases, achieving excellent results, even for acute pain.

Tecar therapy exploits the heat effect, being possible to assume that its biological effect is related to hyperthermia, as used in different forms of thermotherapy. Increased tissue temperature causes arteriolar and capillary dilatation as well as consequent greater blood flow to tissues, leading to higher cellular metabolism and greater tendon and muscle flexibility; in fact, heat improves the contractile performance of muscle.

Low cost and long-term benefits, TT is a useful therapeutic option for the conservative management of pain.

REVIEWER: Create 2.1 Subjects in Materials and Methods.

Basketball players' career, and basketball positions? Add this information.

AUTHORS: We did not consider the basketball positions for this study because the inclusion criteria were very restrictive and, after having analised it, we felt that position does not have influence in our treatment.

REVIEWER: Modify 2.2 Tecar therapy group to Tecar therapy.

AUTHORS: We have changed it in the paper.

REVIEWER: Figure 3 is not clearly visible. It would be good to expand Figure further. It's hard to see what's inside the box.

AUTHORS: We have corrected the size of Figure 3 in the paper (Page 5).

REVIEWER: Describe the Control group more specifically.

AUTHORS: We have added further information in the paper.

REVIEWER: Describe the information about the thermographic assessment tool and Algometer (model M3-20, 20lbf, 10 KGF, 100N), and Leg Motion system (Check 170 Your Motion ® , Albacete, Spain). For example, (Model, Company, City, Nation). Match this part in this manuscript.

AUTHORS: We have added the correspondent information in the paper.

REVIEWER: Describe the discussion in an integrated way, not in each title. Interpret the results of this study by deleting unnecessary content and presenting only necessary parts. Describe the discussion by comparing the results of previous studies.

AUTHORS: We have refreshed it.

Reviewer 3 Report

As attachment

Author Response

Reviewer response 3

AUTHORS: First of all, thank you very much for reviewing this article. We greatly appreciate the effort that this implies, having helped us to improve this paper.

Tittle/ REVIEWER: The title is too big, it could be a little shorter.

AUTHORS: We have changed it.

Abstract / REVIEWER: It is well written, however, it should bring some numerical and statistical results that allow the reader to better understand the study and its results;

AUTHORS: We have added some numerical results in the abstract.

REVIEWER: Some sentences are affirmative and among what was mentioned in the study, it does not allow affirmation but rather to indicate a possibility found in the study;

AUTHORS: We have introduced changes in order to indicate a possibility according to our results.

REVIEWER: Some sentences are affirmative and among what was mentioned in the study, it does not allow affirmation but rather to indicate a possibility found in the study;

AUTHORS: We have changed some of the keywords, now all of them are within the health sciences descriptors.

.

Introduction/ REVIEWER: a) This is very extensive, and on the other hand methodologically explains some points that should this in methodology and not in the introduction;

  1. b) The introduction is not starting from general to specific;
  2. c) It should initially present a more general approach and gradually address the problem (gap) and then present the objective;

AUTHORS: We have refreshed it.

REVIEWER: d) The problem must be better identified to make the study more robust;

AUTHORS: Theoretically, trigger points can lead to restriction in ankle ROM. Within our study, we apply the treatment to increase ankle ROM, it can be explained and further justifed with the literature and relationship with restriction and coactivation of trigger point.

The main objective of the study was analyzed the acute effect of TT on latent MTPs on skin temperature, ankle ROM and VAS in professional basketball players.

REVIEWER: I suggest that you check the journal's standards as some points, such as citations, do not meet the journal's standards should be [1] and not (1), for example, citation 1.

AUTHORS: We have changed it.

Methods/ REVIEWER: The methodology should follow a clearer order of presentation. I suggest that you adopt the following order: drawing, sample, instruments, procedures and statistics to make understanding easier;

AUTHORS: We have changed the presentation.

REVIEWER: The sample should be better explained with the number of subjects presented initially and then present the inclusion and criteria.

AUTHORS: We have refreshed it.

REVIEWER: c) The protocols are not complete, for example, the thermography should go beyond the reference of the equipment used, it should contain the equipment's resolvability, distance, atmospheric conditions and preparation time, and also the position that awaited for the evaluation. The same can be considered for other assessments.

AUTHORS: The protocols of thermography are based on the International Academy of Clinical Thermology. The equipment used was Flir E6, FLIR Systems, Inc, Wilsonville, USA, step for subject position, black background to isolate body temperature.

The temperature range was maintained between 18 and 23ºC. We gave to the subjects different instructions pre-examination.

Prior to imaging, the patient’s body must be given sufficient time to equilibrate with the ambient, therefore they remained 15 minutes sat before the study.

The patient remained standing above the step with 20 cm between their feet. The distance between the camera and step was 3-4 meters depending of the subject. Camera needed to draw a line perpendicular to the point we need to evaluate, for which we used a tripod.

Results/ REVIEWER: Are presented satisfactorily.

AUTHORS: OK, thanks.

Discussion/ REVIEWER: Are presented satisfactorily. I only ask that the limitations of the study be better presented.

AUTHORS: We have improved the limitations ( page 14, line 510 )

Conclusion/ REVIEWER: Are presented satisfactorily. However, practical applications of what was found in the study must be presented.

AUTHORS: The treatment of latent myofascial trigger points using tecar theapy represents an interesting option to resolve muscle function. Even though we are not aware of further physiological benefits, we are sure that this technique does not produce pain after treatment. Also, it is an invasive technique, we can use it with almost everyone and it is easier to apply than other similar techniques.

REVIEWER: Of the 45 references, 21 have been published for more than five years and some are even undated. The formatting of the references is in disagreement with the journal's standards.

AUTHORS: We have now checked and changed it.

REVIEWER: The manuscript presented addresses a relevant research topic.

It would be advisable to do a general review.

Authors' contributions are not in accordance with the journal's norms.

AUTHORS: We have carried out the suggested changes.

Round 2

Reviewer 1 Report

Overall comments:

Thank you for asking me to rereview this paper on the effect of TT on pain, ankle ROM and skin temperature in male basketball players. I still need to be convinced of your overall argument that TT is a useful treatment, given your results show it is no better than sham for pain or ROM and only increases the skin temperature.

Please review your discussion for coherence. A number of paragraphs seem to be covering the same topic and the logical flow of your argument gets lost. Consider synthesising further and shortening your discussion.

Please read very carefully for language editing. I have included a number of places where subtle errors have not been identified but have not spent the time to fully edit the document.

Ensure you replace all ‘subjects’ with ‘participants’

Please include in the paper how you identified Latent TrPs and who identified them.

Specific comments

Line 33: Change to Results: The intervention group….

Line 35 I feel you should really say : There were no differences between groups in the Lunge Test (F[1.68, 53.64]= 2.91, p= 350.072, 2p= 0.08) or pressure algometry (F[3.90] = 0.73, p= 0.539, , 2p = 0.02).

Line 38 Change to: Diathermy can induce changes….

Line 50: Change to ‘injuries related to muscle stiffness.’

Line 57: Suggest changing to: ‘TT is characterized by its speed of action so is used in high performance sports[4] as this tool accelerates the recovery process’

Line 65. 85% of patients with pain do not suffer from MPS! The reference you supplied is of a study of 48 patients with lumbar MPS, not an epidemiological or survey study that ascertains whether chronic pain patients have MFPS. Clarify this sentence or change the reference.

Line 66 fatigability of what? Muscles/ the person?

Line 73: Suggest changing to : In addition, latent MTrPs reduce joint

range of motion due to muscle shortening from muscle and tendon stiffness. (add in the references)

Line 76 The most sensitive palpable nodule where?

Line 82 Please clarify the sentence: ‘MTPs are able to modified muscle contraction’

Line 91 Change to diagnostic imaging

Line 96 change to: physiotherapists may use referred pain patterns to understand and recognize

 how MPS has developed

Line 99 not yet been investigated

Line 106 Change subjects to participants

Line 110 All research staff were blinded..

Line 147 change to: ‘  treated area for 25 minutes

Line 161 Do you mean: To eliminate any cross-effect that may interfere with the results, after the immediately post treatment  measurement, the participants were seated by research staff in a chair , with 90° of hip and knee flexion in a controlled room at a preset temperature.

Line 195 delete ‘scale’ as VAS stands for visual analogue scale.

Line 170 do you mean: The same protocol as the diathermy group was performed, but with the device in off mode (SHAM)

Line 174 do you mean were performed according to the International Academy of Clinical Thermology Guidelines?

Line 195 What were the ‘anchors’ applied to your scale? Was it ‘no pain’ (o) and ‘ maximum pain imaginable (10)? Or other phrases?

Line 203 In the response to reviewers you said you changed this to ‘We have used the highest values of 3 attempts based on of the previous literature’ ensure you have deleted the section ‘

Section 2.4.3 ensure you use past tense in this section

Line 203 Delete ‘3rd or 4th attempt were recorded[34]’ or else you cant take the highest number of the attempts as you didn’t record them according to this sentence.

Line 207 change to ‘ …has been established as <11.5cm etc…’

Line 299 where you have ‘capture’ do you mean ‘time’?

Line 432 change to ‘ the temperature immediately rose by …’

Paragraph from line 439 Include references for the first sentence. I do not understand what is meant by ‘MTPs may adversely affect clinical effects on restricted ROM?? Can you clarify? Should this paragraph and the following 4 paragraphs be synthesised? The logic is difficult to follow between the paragraphs.

Line 448 You say ‘other studies’ but only reference one. Either include another reference or change to ‘one study’

Paragraphs from 474-486 Are you arguing that not finding a difference in pain is an advantage of TT therapy even though you hypothesised a decrease in pain?

Line 486 I don’t think pointing out a 10% decrease is significant since this is only a reduction of 1 point on the VAS and the clinically meaningful change in VAS score is 2.

Line 496 Suggest change to ‘..pressure inhibition treatment reduced pain in MTPs…’

Line 526 Suggest changing to ‘Although thermography is an innovative technique for measuring skin temperature, we are extrapolating that the external temperature reflects the internal; or muscle temperature’.

Line 531: Unsure why you need a larger sample size if you calculated a sample size with enough power to arrive at your conclusions?

Line 531 What other sorts of physiological parameters are relevant?

Line 533 Why would you compare TT to other treatments if you have shown TT has no better effect than doing nothing?

5 I strongly disagree with your conclusion. You did not show an increase in blood vascularization as you did not measure this. You did not reduce pain any more than doing nothing- you are implying an advantageous effect that was not supported by your study. You cannot say that TT will enhance muscle recovery when you cannot say that you are changing anything internally. I still don’t see that your study supports using TT as it was no better than sham except for increasing the skin temperature.

Author Response

REVIEWER REPORT 1:

REVIEWER: Thank you for asking me to rereview this paper on the effect of TT on pain, ankle ROM and skin temperature in male basketball players. I still need to be convinced of your overall argument that TT is a useful treatment, given your results show it is no better than sham for pain or ROM and only increases the skin temperature.

Please review your discussion for coherence. A number of paragraphs seem to be covering the same topic and the logical flow of your argument gets lost. Consider synthesising further and shortening your discussion.

Please read very carefully for language editing. I have included a number of places where subtle errors have not been identified but have not spent the time to fully edit the document.

Ensure you replace all ‘subjects’ with ‘participants’

Please include in the paper how you identified Latent TrPs and who identified them.

AUTHORS: We would like to thank the reviewer for his/her valuable comments, which are really constructive and have definitely helped us to improve the quality of the present manuscript.

REVIEWER: Line 33: Change to Results: The intervention group….

AUTHORS: We have changed it.

REVIEWER: Line 35 I feel you should really say : There were no differences between groups in the Lunge Test (F[1.68, 53.64]= 2.91, p= 350.072, 2p= 0.08) or pressure algometry (F[3.90] = 0.73, p= 0.539, , 2p = 0.02).

AUTHORS: We have refreshed the sentence to clarify.

REVIEWER: Line 38 Change to: Diathermy can induce changes….

AUTHORS: Sorry for the mistake, we have changed the sentence.

REVIEWER: Line 50: Change to ‘injuries related to muscle stiffness.’

AUTHORS: We have changed it.

REVIEWER: Line 57: Suggest changing to: ‘TT is characterized by its speed of action so is used in high performance sports[4] as this tool accelerates the recovery process’

AUTHORS: We have changed the sentence.

REVIEWER: Line 65. 85% of patients with pain do not suffer from MPS! The reference you supplied is of a study of 48 patients with lumbar MPS, not an epidemiological or survey study that ascertains whether chronic pain patients have MFPS. Clarify this sentence or change the reference.

AUTHORS: We have added the reference.

REVIEWER: Line 66 fatigability of what? Muscles/ the person?

AUTHORS: We have refreshed the sentence to clarify.

REVIEWER: Line 73: Suggest changing to : In addition, latent MTrPs reduce joint range of motion due to muscle shortening from muscle and tendon stiffness. (add in the references)

AUTHORS: We have changed the sentence.

REVIEWER: Line 76 The most sensitive palpable nodule where?

AUTHORS: We have refreshed the sentence.

REVIEWER: Line 82 Please clarify the sentence: ‘MTPs are able to modified muscle contraction’

AUTHORS: We have refreshed the sentence to clarify.

REVIEWER: Line 91 Change to diagnostic imaging

AUTHORS: We have changed the sentence.

REVIEWER: Line 96 change to: physiotherapists may use referred pain patterns to understand and recognize how MPS has developed

AUTHORS: We have refreshed the sentence.

REVIEWER: Line 99 not yet been investigated

AUTHORS: We have changed the sentence.

REVIEWER: Line 106 Change subjects to participants

AUTHORS: We have changed it.

REVIEWER: Line 110 All research staff were blinded..

AUTHORS: We have changed it.

REVIEWER: Line 147 change to: ‘ treated area for 25 minutes

AUTHORS: We have refreshed the sentence.

REVIEWER: Line 161 Do you mean: To eliminate any cross-effect that may interfere with the results, after the immediately post treatment measurement, the participants were seated by research staff in a chair, with 90° of hip and knee flexion in a controlled room at a preset temperature.

 AUTHORS: We have changed the sentence.

REVIEWER: Line 195 delete ‘scale’ as VAS stands for visual analogue scale.

AUTHORS: We have changed it.

REVIEWER: Line 170 do you mean: The same protocol as the diathermy group was performed, but with the device in off mode (SHAM)

AUTHORS: We have changed it.

REVIEWER: Line 174 do you mean were performed according to the International Academy of Clinical Thermology Guidelines?

AUTHORS: We have changed the sentence.

REVIEWER: Line 195 What were the ‘anchors’ applied to your scale? Was it ‘no pain’ (o) and ‘maximum pain imaginable (10)? Or other phrases?

AUTHORS: We have changed this part

REVIEWER: Line 203 In the response to reviewers you said you changed this to ‘We have used the highest values of 3 attempts based on of the previous literature’ ensure you have deleted the section ‘

AUTHORS: We have checked it.

REVIEWER: Section 2.4.3 ensure you use past tense in this section

AUTHORS: We have refreshed the sentence to clarify.

REVIEWER: Line 203 Delete ‘3rd or 4th attempt were recorded[34]’ or else you cant take the highest number of the attempts as you didn’t record them according to this sentence.

AUTHORS: We delete it.

REVIEWER: Line 207 change to ‘ …has been established as <11.5cm etc…’

AUTHORS: We have changed the sentence.

REVIEWER: Line 299 where you have ‘capture’ do you mean ‘time’?

AUTHORS: We have changed this part

REVIEWER: Line 432 change to ‘ the temperature immediately rose by …’

AUTHORS: We have changed the sentence.

 REVIEWER: Paragraph from line 439 Include references for the first sentence. I do not understand what is meant by ‘MTPs may adversely affect clinical effects on restricted ROM?? Can you clarify? Should this paragraph and the following 4 paragraphs be synthesised? The logic is difficult to follow between the paragraphs.

 AUTHORS: We have changed the sentence.

REVIEWER: Line 448 You say ‘other studies’ but only reference one. Either include another reference or change to ‘one study’

AUTHORS: We have refreshed the sentence to clarify.

REVIEWER: Paragraphs from 474-486 Are you arguing that not finding a difference in pain is an advantage of TT therapy even though you hypothesised a decrease in pain?

AUTHORS: No, we did not arguing this. We try to explain what happens physiologically, we do not want to imply that the difference in pain reduction is an advantage. In the next paragraph we talk about control group.

REVIEWER: Line 486 I don’t think pointing out a 10% decrease is significant since this is only a reduction of 1 point on the VAS and the clinically meaningful change in VAS score is 2.

 AUTHORS: We have refreshed the sentence to clarify.

REVIEWER: Line 496 Suggest change to ‘..pressure inhibition treatment reduced pain in MTPs…’

 AUTHORS: We have changed the sentence.

REVIEWER: Line 526 Suggest changing to ‘Although thermography is an innovative technique for measuring skin temperature, we are extrapolating that the external temperature reflects the internal; or muscle temperature’.

AUTHORS: We have changed the sentence.

REVIEWER: Line 531: Unsure why you need a larger sample size if you calculated a sample size with enough power to arrive at your conclusions?

AUTHORS: Yes, we calculated the sample size, but in future research we would like to see how the results evolve with other parameters and with more subjects, to determine other accurate details. This does not mean that the sample size of this study was wrong.

REVIEWER: Line 531 What other sorts of physiological parameters are relevant?

AUTHORS: Other sorts could be for example, to investigate through electromyography (intramuscular EMG), ultrasound imaging to see the focal ischemia or hypoxia in the muscle, and other types of test to see if can affect to joint range of motion.

REVIEWER: Line 533 Why would you compare TT to other treatments if you have shown TT has no better effect than doing nothing?

 AUTHORS: We would like to compare in future research with other types of treatments to see what happens and if they behave in the same way as TT.

REVIEWER: I strongly disagree with your conclusion. You did not show an increase in blood vascularization as you did not measure this. You did not reduce pain any more than doing nothing- you are implying an advantageous effect that was not supported by your study. You cannot say that TT will enhance muscle recovery when you cannot say that you are changing anything internally. I still don’t see that your study supports using TT as it was no better than sham except for increasing the skin temperature.

AUTHORS: We have changed the conclusion.

Reviewer 2 Report

This manuscript is revised well. However, manuscript is required English editing service. Please check it.

Author Response

REVIEWER REPORT 2:

This manuscript is revised well. However, manuscript is required English editing service. Please check it.

AUTHORS: We would like to thank the reviewer. We checked this paper.

Reviewer 3 Report

Congratulations to the authors, I consider that the adjustments were made, however, it is necessary to review the formatting of the manuscript, especially in the references

Author Response

REVIEWER REPORT 3:

Congratulations to the authors, I consider that the adjustments were made, however, it is necessary to review the formatting of the manuscript, especially in the references

AUTHORS: We would like to thank the reviewer. We checked this paper.

This manuscript is a resubmission of an earlier submission. The following is a list of the peer review reports and author responses from that submission.